# LIME-Eval: Rethinking Low-light Image Enhancement Evaluation via Object Detection

## Abstract

Due to the nature of enhancement–the absence of paired ground-truth information, high-level vision tasks have been recently employed to evaluate the performance of low-light image enhancement. A widely-used manner is to see how accurately an object detector trained on enhanced low-light images by different candidates can perform with respect to annotated semantic labels. In this paper, we first demonstrate that the mentioned approach is generally prone to overfitting, and thus diminishes its measurement reliability. In search of a proper evaluation metric, we propose LIME-Bench, the first online benchmark platform designed to collect human preferences for low-light enhancement, providing a valuable dataset for validating the correlation between human perception and automated evaluation metrics. We then customize LIME-Eval, a novel evaluation framework that utilizes detectors pre-trained on standard-lighting datasets without object annotations, to judge the quality of enhanced images. By adopting an energy-based strategy to assess the accuracy of output confidence maps, our LIME-Eval can simultaneously bypass biases associated with retraining detectors and circumvent the reliance on annotations for dim images. Comprehensive experiments are provided to reveal the effectiveness of our LIME-Eval. Our code will be made publicly available.

## 1 Introduction

Low-light conditions significantly challenge imaging by reducing the visibility of important details and/or introducing distortions in captured images, such as noise, blur, and color shifts. The poor quality of images captured in such conditions not only hampers everyday photography but also poses serious issues in fields where image clarity is critical, such as surveillance, navigation, and astrophotography. Consequently, low-light image enhancement has emerged as an essential technique to improve the quality of images taken in unsatisfactory lighting environments.

The objective of low-light image enhancement is to concurrently brighten dark regions, enhance suppressed details, preserve color fidelity, and eliminate potential artifacts. In other words, it is desired to generate high-quality images that closely resemble those taken under "good"[1] lighting conditions. While substantial progress has been made in this domain, spanning from histogram equalization (Trahanias & Venetsanopoulos, 1992) to advanced deep-learning approaches (Cai et al., 2023a; Zhang et al., 2019; Guo & Hu, 2023), a key challenge persists in objectively evaluating the performance of enhancement algorithms. Historically, the image quality assessment (IQA) of enhancement has depended on reference-based metrics (*e.g.*, PSNR, and SSIM), which compare the enhanced results to a reference image deemed to be of high quality.

We argue that reference-based metrics are unsuitable for low-light **enhancement**, because the very nature of the problem precludes the existence of reliable reference images. On the one hand, capturing such references with identical settings, except for proper illumination, is inherently challenging. On the other hand, even if reference images are obtained under well-controlled settings, many/infinite variations of "well-lit" conditions exist, making it hard to determine which specific scenario aligns with the "best". This absence of a definitive standard (actually for all enhancement tasks) complicates the evaluation process and necessitates the development of no-reference assessments. One might suggest using no-reference IQA metrics, such as NIQE (Mittal et al., 2013a) and

---

[1]As an enhancement task, there is no well-defined optimal lighting condition.

BRISQUE (Mittal et al., 2012b), which are currently mainstream. However, assessing the quality of enhanced images involves a complex interplay of quality and aesthetic issues (color restoration and lightness). Existing no-reference IQA metrics are often inconsistent or even contradictory with human perception, rendering them unreliable in low-light scenarios (Saha et al., 2023; Guo & Hu, 2023), as we will demonstrate in Sec. 4.

Alternatively, the community has begun to reassess the effectiveness of low-light image enhancement techniques by examining their impact on downstream vision tasks, particularly object detection. The core idea is that the performance of downstream models can serve as a proxy for human perception. If objects within an image are adequately illuminated, they should be easily identifiable by both humans and machines. A widely adopted evaluation protocol in recent studies requires fine-tuning an object detector on images enhanced by different methods (Cui et al., 2022; Cai et al., 2023b; Zhou et al., 2024). However, this retraining process raises a significant concern: *i.e.*, overfitting. The detector may become overly tailored to the specific characteristics of the enhanced images, disregarding its resemblance to a natural, well-illuminated image. This leads to a fundamental question: *Is fine-tuning detectors a valid approach to evaluating enhancers?* In Sec. 4, we will demonstrate that the performance of fine-tuned detectors does not necessarily correlate with the quality of enhanced images. With the application of an appropriate augmentation strategy, fine-tuning inadequately enhanced images can still yield better detection performance than even the most advanced enhancement methods. This finding indicates that the fine-tuning protocol conflates the effectiveness of the enhancement algorithms with the adaptability of detection models, ultimately compromising the fairness and reliability of the evaluation.

To remedy the aforementioned flaw, a straightforward strategy shall deploy detectors pre-trained on data captured under normal-lighting conditions to evaluate enhanced images, using annotations from the original low-light images (Wang et al., 2021; Ma et al., 2022a). The underlying premise is that the closer the enhanced results resemble the normal-lit image domain, the better the detection performance will be. while this manner takes advantage of the inherent generalizability of models trained under standard illumination to assess the fidelity of low-light enhancements, it introduces its own set of challenges related to semantic labels. For one thing, obtaining accurate annotations for low-light images is more difficult and time-consuming than for those captured under normal lighting conditions, as the reduced visibility and contrast in low-light images increase ambiguities in object boundaries and classifications. For another thing, the reliance on annotated labels restricts the flexibility of evaluation. Moreover, annotating a large dataset of low-light images to establish a reliable benchmark for evaluation is labor-intensive, limiting the scalability of this approach.

In this work, we introduce the first online benchmark platform, *LIME-Bench*, designed to collect human preferences for assessing low-light enhancement methods. Through the data collected, we verify that although less sensitive to color shift, directly applying pre-trained detectors serves as an effective critique for evaluating low-light image quality than a series of previously applied IQA methods, offering a decent proxy for evaluation. We then propose *LIME-Eval*, a label-free evaluation protocol for evaluating low-light image enhancers. Grounded in a pioneering energy-based criterion, our method sidesteps the biases and time-consuming processes associated with retraining detectors while liberating the demand for both reference images and detection labels. These features broaden the applicability of LIME-Eval to unlabeled and reference-free low-light scenarios.

Our primary contributions are summarized as follows:

- By retraining detectors on enhanced images produced by various low-light enhancement methods, we find that, under appropriate data augmentation conditions, higher detection accuracy does NOT necessarily correlate with superior enhancement quality.

- We collect 6,362 feedback pairs from 750 users, encompassing factors such as blurriness, exposure, noisiness, color, and overall quality across 14 low-light enhancement methods to construct the first low-light user preference dataset, LIME-Bench. Utilizing this data, we benchmark non-reference image quality assessment methods in prior arts and validate the correlation between detector-based evaluations and human preferences.

- We introduce a novel energy-based evaluation framework, say LIME-Eval, which effectively links the quality of enhanced images with the performance of object detection without object labels or reference images. Comprehensive experimental results and analyses confirm LIME-Eval's effectiveness and evidence of its potential to guide low-light enhancers.

## 2 RELATED WORK

**Low-light Image Enhancement** aims to tackle multiple degradations present in dark images such as noise, low contrast, and color shift. Early methods, like histogram equalization, sought to improve image visibility by adjusting global and/or local contrast. The advent of deep learning has led to innovative approaches. Within this context, Retinex theory–which conceptualizes an image as the product of reflectance and illumination components–has gained significant attention. Several schemes based on this paradigm endeavor to produce normal-light images by modulating the illumination component and estimating reflectance (Fu et al., 2016; Ng & Wang, 2011). In advancing the exploration of attention mechanisms, the transformer architecture integrates self-attention with convolutional processes to simultaneously extract long/short-range dependencies. As a representative work, Retinexformer (Cai et al., 2023b) introduces a self-attention module, based on retinex theory and transformer architecture. Focusing on the illumination of an image, an illumination adaptive transformer (IAT) was proposed (Cui et al., 2022), notable for its minimalist design of just 90k parameters and its efficiency in addressing illumination adjustments. Guo and Hu (Guo & Hu, 2023) decoupled the entanglement of noise and color distortion, further alleviating the challenges of low-light enhancement in the presence of complex degradations. In the absence of ground truth, Guo *et al.* (Guo et al., 2020) proposed an unsupervised method adjusting the illumination with LE-curve, achieving reasonable results at an impressively fast pace. These developments represent a significant leap in low-light image enhancement. However, as previously discussed, the lack of exact reference images for enhancement tasks necessitates further research to explore methods for assessing enhanced images in reference-free fashions.

**Image Quality/Aesthetic Assessment** has always been a fundamental task in image processing, especially in enhancement, compression, and restoration. Traditional methods heavily rely on full-reference metrics, with Peak Signal-to-Noise Ratio (PSNR) and Structural Similarity Index (SSIM) as two prominent examples, comparing processed results against high-quality references. However, it is hard to capture decent reference images in many tasks, especially for those enhancement tasks, that have no ground truth by its definition. This discrepancy has spurred the development of no-reference image quality assessment (NR-IQA) methods, which forgo the need for references. Early NR-IQA research primarily focused on specific distortions, notably JPEG compression (Wang et al., 2002; Marziliano et al., 2004). The introduction of the LIVE dataset (Sheikh et al., 2006) marked a shift toward general-purpose NR-IQA, which leverages natural scene statistics (NSS) from spatial (Mittal et al., 2012a; 2013b) or transform domains (Moorthy & Bovik, 2011; Saad et al., 2012) to assess image quality, predicated on the premise that deviations from the statistical regularities found in natural images correlate with perceived visual quality (Simoncelli & Olshausen, 2001). With the expansion of IQA datasets and the growing influx of images, deep learning has emerged as the predominant force. To compensate for the shortage of manually-labeled data, strategies like patchwise training (Bosse et al., 2018; Kang et al., 2014), transfer learning (Zeng et al., 2018), and quality-aware pre-training (Ma et al., 2018; Liu et al., 2017) have been developed. Up-to-date NR-IQA research cooperated with innovations like active learning, meta-learning, patch-to-picture mapping, loss normalization, and adaptive convolution. These advancements aim to enhance generalizability (Wang & Ma, 2022), enable rapid adaptation (Zhu et al., 2020), improve local quality prediction (Ying et al., 2020), expedite convergence (Li et al., 2020), and facilitate content-aware quality assessment (Su et al., 2020). These IQA methods have demonstrated significant success in evaluating the quality of enhanced images. Nevertheless, there remains a noticeable gap in understanding the relationship between these quality evaluations and the performance of subsequent downstream tasks. Bridging this gap is crucial for a more comprehensive assessment of image enhancement techniques in real-world applications.

**Benchmarking Low-light Enhancement with Detection** is tough due to the subjective nature of image quality and the lack of suitable standards for comparison. This has led researchers to explore alternative evaluation strategies, *e.g.*, subjective assessment by human observers, or the use of synthetic datasets where ground truth is artificially generated. The ExDark dataset (Loh & Chan, 2019) serves as a repository of low-light object images, defined by criteria including low illumination levels or pronounced variations in lighting. The DarkFace dataset (Chen et al., 2018) offers a collection of low-light images with face annotation. Despite the value of these datasets, they suffer from limitations in scalability and often fail to capture the complexity of real-world scenarios. Thus, it is imperative to develop innovative methodologies for evaluating enhanced images in the absence of reference images.

Table 1: Quantitative comparison in detection accuracy mAP on the test split of ExDark. The best and second-best results by each scheme are in **bold** and underlined, respectively. The results in the 'Dim image' column are obtained by directly training the detector on images without enhancement.

| | Dim image | Zero-DCE | Bread | Bread-round | IAT | RetinexFormer |
|---|---|---|---|---|---|---|
| Enhance First | N/A | 45.9 | 45.9 | N/A | **47.6** | 46.1 |
| Augmentation First | **49.2** | 49.2 | 48.9 | 48.8 | 49.0 | 48.5 |
| Direct Eval | 35.0 | 34.0 | **35.4** | N/A | 35.3 | 34.2 |

(a) Input  (b) Zero-DCE  (c) Bread  (d) IAT  (e) RetinexFormer

Figure 1: Qualitative comparisons on samples from the ExDark dataset. Please zoom in for more details. Due to the page limit, more cases can be found in the appendix.

**Detection in Low-light Scenarios.** Low-light environments pose challenges for image detection, prompting research into three main approaches: 1) detection-specific enhancement (Sun et al., 2022; Hashmi et al., 2023), 2) improved low-light enhancement for detection (Ma et al., 2022a; Guo et al., 2020; Li et al., 2024; Cui et al., 2022), and 3) optimized detector training (Cui et al., 2021; Cui & Harada, 2024). Detection-oriented enhancers (Sun et al., 2022; Hashmi et al., 2023) improve detection but often fail to enhance visual quality or generalize across detectors. The second category focuses on balancing visual appeal with detection accuracy, which aims to produce images that are aesthetically pleasing while ensuring they are optimized for downstream machine vision tasks (Guo et al., 2020; Li et al., 2024). The third approach optimizes detector training for low-light conditions, using techniques like domain adaptation (Dai & Gool, 2018; Wang et al., 2021; Du et al., 2024) and multi-task learning (Cui et al., 2021; Cui & Harada, 2024). Our work explores the link between human and machine vision, with our LIME-Eval serving both as an enhancement evaluator and a potential means to improve detection performance.

**Energy-based Models** (EBMs) are versatile, non-normalized probabilistic models introduced by (LeCun et al., 2006). They define relationships among variables by assigning a scalar energy value to each multivariate instance. Unconstrained by the need to maintain normalized probabilities, EBMs have found application across a wide array of tasks (Li et al., 2022; Du et al., 2020; 2022). Thanks to their ability to represent complex, high-dimensional data distributions, EBMs have also been applied in generative modeling tasks (Arbel et al., 2021). The work (Grathwohl et al., 2020) demonstrates how classifiers can inherently function as EBMs, further broadening their applicability. This perspective on energy has been harnessed for tasks such as out-of-distribution detection (Lafon et al., 2023) and automated evaluation of classification models (Peng et al., 2024). Inspired by these advancements, our approach adopts energy-based statistics as a proxy for average accuracy, showcasing the model's adaptability and effectiveness in evaluation contexts.

## 3 RETHINKING EVALUATION PROTOCOL

The information bottleneck theory (Tishby et al., 2000) suggests that neural network operations can result in information loss, potentially obscuring critical clues for high-level tasks. This understanding has influenced the evaluation of low-light enhancement methods, where performance is often assessed by retraining downstream recognition models on images enhanced by these enhancers. But, the validity of such an evaluation scheme is questionable. To manifest this, we carefully select four low-light enhancement techniques that exemplify the diversity in current low-light enhancement approaches, including Zero-DCE (Guo et al., 2020), Bread (Guo & Hu, 2023), IAT (Cui et al., 2022) and RetinexFormer (Cai et al., 2023b).

Without loss of generality, we initiate our investigation with object detection on the ExDark (Loh & Chan, 2019) dataset, a widely recognized benchmark for low-light conditions. As for our base detector, we choose the medium version of YOLOX (Ge et al., 2021). All the involved enhancers, pre-trained on the LOL (Chen et al., 2018) dataset, remain fixed during detector training. We experiment with two distinct training settings: 1) *Enhancement First Augmentation After*: Low-light images are enhanced first and saved in a lossless format, strong augmentation techniques (*e.g.*, Mosaic, shear, mixup); and 2) *Augmentation First Enhancement After*: Augmentation techniques are applied directly to the original low-light images, enhancement is performed on-the-fly, applied to the augmented output.

Referencing Tab. 1, it is noteworthy that while Bread exhibits superior visual results (please see Fig. 1 for visual comparisons), it does not achieve the highest detection performance under either of the tested settings. Interestingly, the detection performance of Bread (45.9), IAT (47.6), and RetinexFormer (46.1), when enhanced before data augmentation, does not surpass dim (low-light) images (49.2). In contrast, when adopting the augmentation-first strategy, all the methods show a marked improvement in performance. To ensure that performance gains is not attributable to information loss during the process of saving continous enhancement result into discrete images, we conducted an experiment with Bread by clipping and quantizing its intermediate output to 8 bit integers before detector training. The modified version, referred to as Bread-round, demonstrated performance comparable to the original Bread. This indicates that the performance drop in the enhancement-first setting is inherent to the enhancement-first scheme itself.

This evidence highlights a key limitation of the retraining: it encourages detectors to optimize the utilization of available input clues and adapt specifically to the enhanced input domain. Under this paradigm, the training process compels the detector to rely solely on these input clues, rather than leveraging the common sense that underpins human perception. To tackle the overfitting issue, a straightforward solution is to forego training the detector, and instead use models trained on large-scale normal-light datasets for direct inference on low-light images. The results are detailed in Tab. 1, under "Direct Eval". The findings reflect that Bread and IAT, which visually resemble normal-light images more closely, outperform those without enhancement and models Zero-DCE (which suffers from poor noise suppression) and RetinexFormer (which introduces artifacts due to overfitting). As the misalignment between the focus of fine-tuned detectors and actual perceptual quality results in skewed evaluation fairness, direct evaluation using pre-trained detectors without additional fine-tuning seems to offer a more unbiased protocol for assessing low-light enhancement methods.

# 4 BENCHMARKING MACHINE-HUMAN CONSENSUS VIA LIME-BENCH

However, the direct evaluation approach raises new questions that warrant further investigation: *Is there always a consensus between human preference and direct detection performance?* and *If discrepancies arise, which factors have the most significant impact?*

To address these questions, we conducted user studies using the images from the ExDark dataset (Loh & Chan, 2019), comparing dim images with outputs from 14 low-light enhancement methods. These methods include one optimization-based approach (LIME (Guo et al., 2017)), 7 supervised methods (Bread (Guo & Hu, 2023), Kind (Zhang et al., 2019), Retinexformer (Cai et al., 2023b), IAT (Cui et al., 2022), SNR (Xu et al., 2022), LLFlow (Wang et al., 2022) and PyDiff (Zhou et al., 2023)), and 6 unsupervised methods (QuadPrior (Wang et al.), LightenDiffusion (Jiang et al., 2024), SCI (Ma et al., 2022b), ZeroDCE (Guo et al., 2020) PairLIE (Fu et al., 2023) and NeRCo (Yang et al., 2023) ). Inspired by Chatbot Arena (Chiang et al., 2024), we presented users with pairs of images generated by different enhancement methods and randomly selected one of five aspects—overall quality, illumination, noise/artifacts, blurriness, or color. Participants were then asked to choose the better option based on the selected criterion. To quantify user preferences across all methods, we employed the Elo rating system to convert these pairwise comparisons into a comprehensive rating. Further details of the study can be found in the appendix.

As depicted in Fig 2 (a), the direct evaluation of detection performance shows a strong correlation with user preferences from the study. The Spearman correlation coefficient $r$ is 0.703, indicating a robust positive relationship that suggests a general consistency between detection scores and user-assigned image ratings across different methods. This correlation is statistically significant, with a

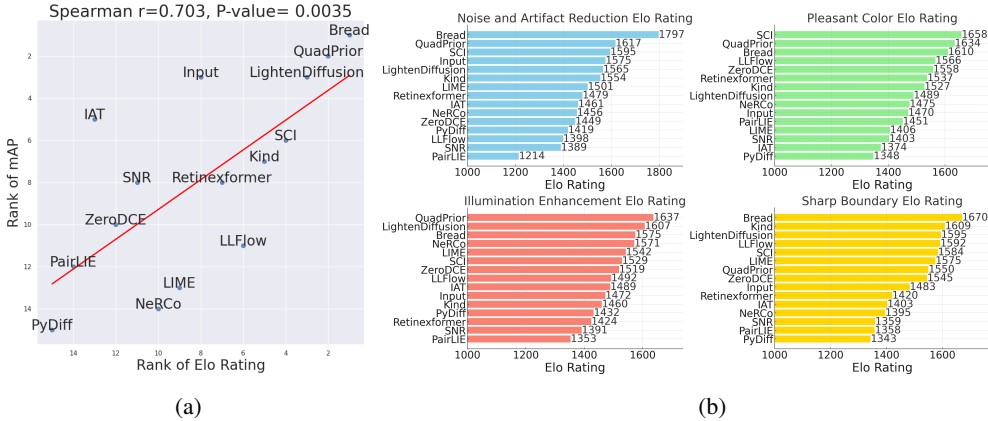

Figure 2: User preference study. (a) plots the rank of the overall user preference (Elo Rating) in relation to detection performance (mAP). (b) depicts the Elo Ratings respectively for noise/artifact reduction, illumination enhancement, color restoration, and boundary sharpness.

p-value of 0.0035. However, several outliers are evident in the figure: 1) While IAT demonstrates strong detection performance, it ranks 13th in overall user preference. As shown in Fig. 2(b), IAT's color rendering is particularly unappealing, a crucial factor to human perception that may be overlooked by detectors trained with extensive color jitter augmentation. 2) The input image ranks 8th in the Elo rating but achieves third-best detection performance. As illustrated in Fig. 2(b), although the dim image is poorly illuminated, its noise and JPEG artifacts are (of course) less noticeable in the darker areas. In contrast, some enhancement methods (*e.g.*, NeRCo) may inadvertently introduce additional artifacts during the enhancement process, resulting in lower detection performance despite potential improvement in illumination.

Consequently, direct evaluation with detectors is less sensitive to color shift and poor illumination but is more sensitive to noise and artifacts. Despite these discrepancies, the overall correlation indicates that direct detection performance can serve as a reasonable proxy for assessing enhancement quality. To illustrate the advantage of using direct evaluation scheme, we selected 6 popular IQA/IAA methods in low-light enhancement, including NIQE (Mittal et al., 2013a), BRISQUE (Mittal et al., 2012b), MUSIQ (Saha et al., 2023), ClipIQA (Wang et al., 2023), NIMA (Esfandarani & Milanfar, 2018) and LIQE (Zhang et al., 2023). We fed the same input images used in the user study for benchmarking. The results can be found in Fig. 3. BRISQUE, ClipIQA, LIQE, and MUSIQ tend to favor the outputs from PyDiff and NeRCo while overlooking the performance of LightenDiffusion. Among these methods, NIMA reaches the best correlation with human preferences, with a Spearman $r$ of 0.457. However, the alignment between these quality assessment approaches and human perception remains inferior to that of the detection-based evaluation, highlighting the reliability of direct detection performance as an assessment metric.

## 5 TOWARDS LABEL-FREE EVALUATION THROUGH LIME-EVAL

Previous experimental validation still relies on annotated datasets. The dependency on annotated labels presents a series of significant hurdles. Primarily, securing precise annotations for low-light images is notably more difficult and time-consuming than for well-lit images due to the inherent challenges associated with low visibility. The reduced contrast and clarity in low-light conditions often lead to unclear object boundaries and categories, elevating the potential for inaccuracies in annotations. Furthermore, the endeavor to annotate a large-scale dataset of low-light images for establishing a dependable benchmark demands considerable labor and expense, thereby constraining the scalability and practicality of such evaluative methods.

Given the reliance on annotated datasets in these experiments, there emerges a pressing need for an evaluation methodology that operates independently of labels. A label-independent evaluation approach would streamline the assessment of low-light image enhancement techniques and broaden the applicability across diverse and unlabeled datasets. Consequently, exploring and developing an evaluation strategy that transcends the need for annotated datasets becomes a critical next step

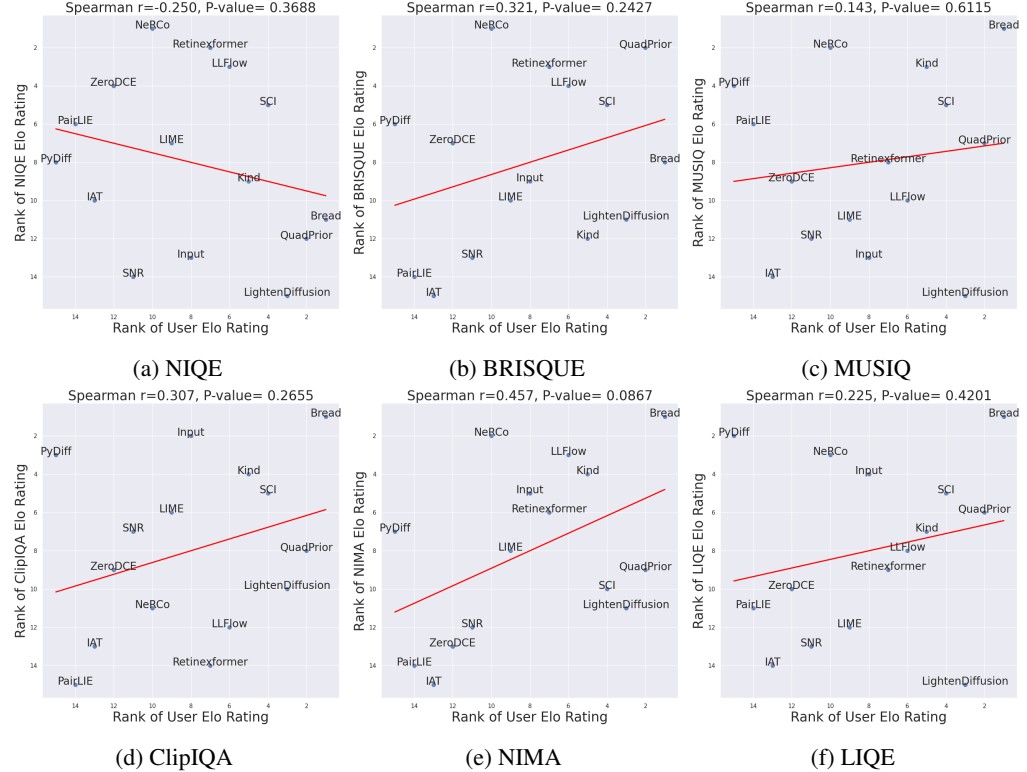

Figure 3: Correlations between user preference and popular IQA approaches.

in advancing the field of low-light image enhancement. In what follows, we shall introduce our LIME-Eval, a label-free evaluation metric, as a pioneering exploration of this problem.

## 5.1 DETECTION-ORIENTED ENERGY-BASED MODELING

Recent studies (Grathwohl et al., 2020; Peng et al., 2024) have demonstrated that classifiers can be interpreted as Energy-Based Models (LeCun et al., 2006) (EBMs), exhibiting an intuitive property: correctly classified samples are associated with lower energy values, whereas misclassified ones are assigned higher energies. Leveraging this insight, we employ energy-based modeling to evaluate the accuracy of object detection systems. It can map data point $x$ with any dimension into a scalar through an energy function $Z(x) : \mathbb{R}^D \rightarrow \mathbb{R}$. To transfer the energy function into a probability density function $p(x)$, one could adapt the Gibbs distribution as follows:

$$p(y \mid x) = \frac{e^{-Z(x,y)/T}}{\int_{y'} e^{-Z(x,y')/T}} = \frac{e^{-Z(x,y)/T}}{e^{-Z(x)/T}}, \tag{1}$$

where $\int_{y'} e^{-Z(x,y')/T}$ is the partition function by marginalizing over label $y$, and $T$ is a positive temperature constant. Now the Gibbs free energy $Z(x)$ at the data point $x$ with the negative of the log partition function can be written as:

$$Z(x) = -T \cdot \log \int_{y'} e^{-Z(x,y')/T}. \tag{2}$$

Consider a $K$-category classifier $f$, which maps input vector $x$ into $K$ logits, with the softmax function, we can parameterize a categorical distribution via:

$$p(y \mid x) = \frac{e^{f_y(x)/T}}{\sum_{k=1}^{K} e^{f_k(x)/T}}, \tag{3}$$

where $f_y(\cdot)$ denotes logit corresponding to $y$-th term of $f(x)$. Thus, the energy function can be expressed as:

$$Z(x) = -T \cdot \log \sum_{j=k}^{K} e^{f_k(x)/T}. \tag{4}$$

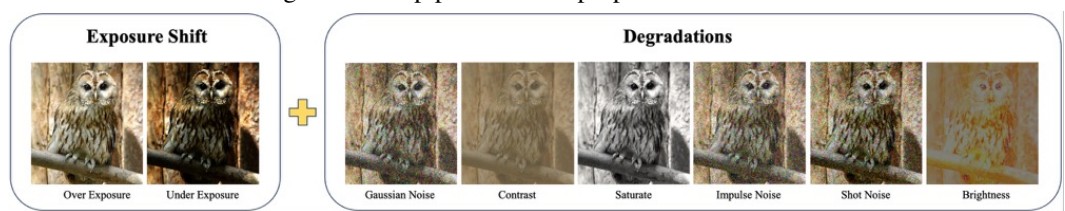

Figure 4: The pipeline of our proposed LIME-Eval.

Figure 5: A visualization of synthetic setting. More details can be found in Appendix.

The above modeling can only be applied to the task of classification, which has been adapted to the AutoEval (Peng et al., 2024). The classification task only involved one overall prediction. To adopt it into our target task, *i.e.* object detection[2], which contains both classification output $x_{cls} \in \mathbb{R}^{K \times H \times W}$ and object output $x_{bg} \in \mathbb{R}^{H \times W}$. Within the detection pipeline, the detector determines whether a data point corresponds to a valid object via background classification score $x_{bg} \in [0, 1]$. A higher score indicates greater confidence in treating the data point as part of an object.

In an object detector, we observe that the correctly classified areas are sparse, only with a minimum difference among clear and damaged images, but the logits in those less confident areas are more sensitive to degradations. To address these areas, we propose to amplifies the influence of the classification score in these areas. Specifically, we apply the formulation $(1 - x_{bg}) \cdot x_{cls}^y$, where $x_{bg}$ represents the background score and $x_{cls}$ denotes the classification score. As this adjustment reduces the overall score magnitude, we introduce a square-root transformation to balance the scaling, yielding a refined confidence measure expressed as:

$$x_r^y = \sqrt{x_{cls}^y \cdot (1 - x_{bg})}, \tag{5}$$

where $x_r^y$ denotes the $y$-th logit of $x_r$. After that, we integrate Eq. (4) into final evaluation function $E(x_{cls}, x_{bg})$ as follows:

$$E(x_{cls}, x_{bg}) = -\sum_{i,j}^{H,W} T \log(\sum_y e^{x_r^{y,i,j}/T}), \tag{6}$$

where $x_r^{y,i,j}$ denotes logit corresponding to $y$-th term of $x_r$ at $(i, j)$. This indicator transforms spatial confidence information into a distribution measure, which can be further aggregated over the dataset for a dataset-level metric. The overall pipeline of our LIME-Eval is illustrated in Fig. 4. After extracting background prediction $x_{bg}$ and classification prediction $x_{cls}$, the two feature maps are fused via $\sqrt{x_{cls}(1 - x_{bg})}$. The energy is calculated as in Eq. (6). Finally, we identify the image with the lowest energy value as the optimal one.

## 5.2 EXPERIMENTAL VALIDATION

**Correlation Studies on Synthesised Datasets.** Having the evaluation function defined, it becomes feasible to assess images without relying on labels. To show that our proposed energy metric aligns with detection performance, we synthesized images from the validation set of MS-COCO (Lin et al., 2015) with low-light-related distortion. To obtain a more accurate approximation of the mean Average Precision (mAP), we begin by pre-calibrating the images using a synthetic dataset derived from the validation split of the MS-COCO dataset. Inspired by typical low-light enhancers (Guo & Hu,

---

[2]For simplicity, here we omit the multi-scale outputs and consider the output as heatmaps in $H \times W$.

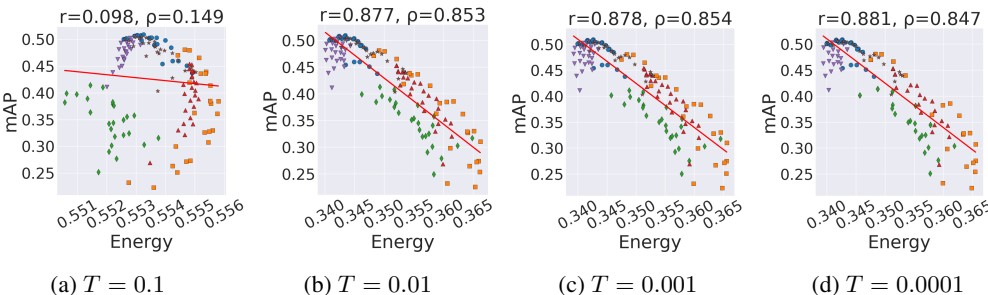

Figure 6: Energy versus mAP under different temperatures on our synthesized dataset. Data points as ○, □, ◇, △,▽,⋆ refers to over-smooth, Gaussian Noise, impulse noise, shot noise, brightness adjustment and saturate adjustment. The calibrated energy function is plotted in red line.

Table 2: Spearman correlation comparison between quality/aesthetic/lightness assessment and our LIME-Eval. The best is in **bold** and the second-best is underlined.

|   | BRISQUE | NIQE | LIQE | MUSIQ | ClipIQA | NIMA | LOE | DeT | Ours |
|---|---------|------|------|-------|---------|------|-----|-----|------|
| $r$ | 0.321 | -0.250 | 0.225 | 0.143 | 0.307 | 0.457 | -0.11 | 0.300 | **0.593** |

2023; Guo et al., 2017), which first perform exposure correction and then handle degradation, our synthesis pipeline is similarly divided into two primary phases to address two types of distortions: 1) *Exposure Shift*. The first phase focuses on the prevalent issues of over/under-exposure commonly observed in low-light enhancement outputs. We employ gamma correction to simulate them; and 2) *Degradation*. The second phase replicates degradations such as ineffective noise suppression, leading to either noise persistence or excessive image smoothing, as well as color distortions. To simulate these effects, we introduce impulse, shot, and Gaussian noises for the former, and employ Gaussian blur for the latter. Further, we adopt strategies to reduce saturation and brightness, mimicking color distortions.

A detailed depiction of the synthesizing process is illustrated in Fig. 5. To quantitatively evaluate the performance of our energy metric, we employed two statistical measures: Pearson's correlation coefficient ($\rho$) and Spearman's rank correlation coefficient ($r$). Pearson's correlation ($\rho$) measures the linear relationship between the energy metric values and detection performance scores, providing insight into how well the metric predicts actual performance improvements. Spearman's rank correlation ($r$), on the other hand, assesses how well the relationship between the energy metric and detection performance follows a monotonic function. This is particularly useful for understanding the metric's ability to rank enhancement methods according to their impact on detection performance. A YOLOX-x model trained on the MS-COCO dataset is adopted as $f$ (aforementioned classifier). As can be seen from Fig. 6, our proposed method shows a strong correlation with mAP ($r = 0.881$, $\rho = 0.847$), indicating that the proposed evaluation function aligns closely with actual mAP, even without the help of labels.

**Consensus with Human-preference.** We also employ user preference from LIME-Bench to benchmark the performance of the proposed method. Our competitors consist of image quality assessment methods (BRISQUE (Mittal et al., 2012b), NIQE (Mittal et al., 2013a), LIQE (Zhang et al., 2023), MUSIQ (Ke et al., 2021), and ClipIQA (Wang et al., 2023)), an image aesthetic assessment method NIMA (Esfandarani & Milanfar, 2018), a color assessment method DeT (He et al., 2023), and a lightness assessment method LOE (Wang et al., 2013). As reported in Tab. 2, our method exhibits a good correlation with human preferences compared to its competitors, demonstrating its alignment with perceptual quality judgments, even if it is not trained on any image-quality-related dataset.

## 5.3 BACKPROPAGATION OF ENERGY HELPS ENHANCERS

Since our energy function is differentiable, it can perform as a loss function to provide additional regularization for low-light enhancers. We verify this by integrating our energy function into the training process of Retinexformer. To make a comprehensive analysis, we use not only reference-based metrics (LPIPS (Zhang et al., 2018) and SSIM (Wang et al., 2004)), but also detectors. Our LIME-Eval uses YOLOX (Ge et al., 2021) as the base detector. When testing on the YOLOX

Table 3: Backpropagation Analysis. For the "RF + Lime-Eval" variant, we retrained Retinexformer on the LOL-v2 dataset and added LIME-eval as an additional loss function.

| Method | Referenced | | Traditional Detector | | | Open-vocabulary Detector | | |
|---|---|---|---|---|---|---|---|---|
| | LPIPS | SSIM | mAP | AP50 | Recall | mAP | AP50 | Recall |
| RF | 0.186 | 0.834 | 29.5 | 55.8 | 48.6 | 35.1 | 63.2 | 56.7 |
| RF + LIME-Eval | 0.162 | 0.846 | 29.5 | 55.9 | 49.0 | 35.2 | 63.3 | 57.4 |

(a) YOLOX-Tiny     (b) YOLOX-S     (c) YOLOX-L     (d) YOLOX-X

Figure 7: Energy versus mAP under different base detectors on our synthesized dataset.

detector, we obtain gains in mAP (from 34.0 to 34.7) and AP50 (from 64.2 to 64.9). To validate the performance on other architectures, we adopt another traditional detector YOLOv8 (Ultralytics, 2023) and an open-vocabulary detector YOLO-World (Cheng et al., 2024) for a direct evaluation. As reported in Tab. 3, the model armed with our energy function enjoys favorable gains in both low-level metrics (LPIPS and SSIM) and detection metrics (mAP, AP50, and Recall) from two different paradigms of detectors, especially the recall. Please note that the performance gain comes from the backpropagation of energy, without the help of ground-truth labels or reference images. These results indicate that the energy function contributes effectively to regularization, aiding the enhancement process. Visual comparisons can be found in the appendix.

### 5.3.1 ABLATION STUDY

**The Effect of Hyper-parameter $T$.** Given that our framework relies on a single hyper-parameter, $T$, we have conducted a series of experiments to assess its sensitivity and impact on performance. The experimental results are systematically presented in Fig. 6. The findings show that our energy-based metric maintains a strong correlation across a range of values from 0.01 to 0.0001, indicating that the method stays stable over a broad spectrum of temperature settings.

**The Effect of Model Size.** We also explored the impact of model size on the performance of our framework by experimenting with different versions of the YOLOX architecture: YOLOX-Tiny, YOLOX-s, YOLOX-l, and YOLOX-x. This investigation aims to understand how the size of the base detector influences detection accuracy, processing speed, and overall system efficiency within our enhanced low-light image evaluation setup. The outcomes of these experiments, which detail the trade-offs associated with each model size, are documented in Fig. 7. As we can observe from the figure, the larger the model, the stronger the correlation energy with the mAP will be. When we scale the model back to YOLOX-Tiny, the connection between energy and mAP vanishes.

## 6 CONCLUSION AND FUTURE WORK

In this study, we present a comprehensive evaluation of low-light image enhancement techniques from the aspect of object detection. Our analysis highlights the limitations of retraining recognition models on enhanced images, which often leads to overfitting and undermines the fairness and accuracy of evaluations. By leveraging an energy-based evaluation framework, we propose a novel approach that mitigates and provides a robust and equitable assessment of enhancement techniques. The results demonstrate that this method is an attractive solution for reflecting the performance of low-light enhancement algorithms. Beyond low-light enhancement, the findings underscore the potential of energy-based models to serve as a versatile tool for assessing diverse image processing tasks. Future work will explore the broader applicability of this framework to similar tasks in image enhancement and restoration.

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

## A  LIME-BENCH DETAILS

Our LIME-bench collects data through an online user survey. A screenshot of our system can be found in Fig. Specifically, we randomly select an image, a particular attribute, and two enhancement methods (including the original input). Users are then asked to choose between four options: Image 1 is better, Image 2 is better, both are good, or both are bad. We adopt the Elo rating system to obtain the final rating for each method. For a pair of user preference, if the user can tell which one is better, then we update the score with $k$ set to 16. However, when the user voted for "both are good/well", we treated it as the two competitors both win/lose from the original input, since this requires 2 times of score update, we down-weighted the $k$ to 8 in this situation.

### A.1  IMPLEMENTATION DETAILS

In this work, we use PyTorch to implement our LIME-eval framework. All of our experiments are carried out on NVIDIA RTX3090 GPUs. The detectors and enhancers are trained with the code and configuration (optimizer, learning rate random seeds, etc.) provided by the authors to provide a best-effort fair comparison, except for Tab. 1, where we carefully tuned the parameters for the best performance since no existing training recipe for us to follow.

### A.2  DATA SYNTHESIS

The data synthesis pipeline we have used comprises two types of distortions, the settings of which are as follows:

1. Exposure Shifts

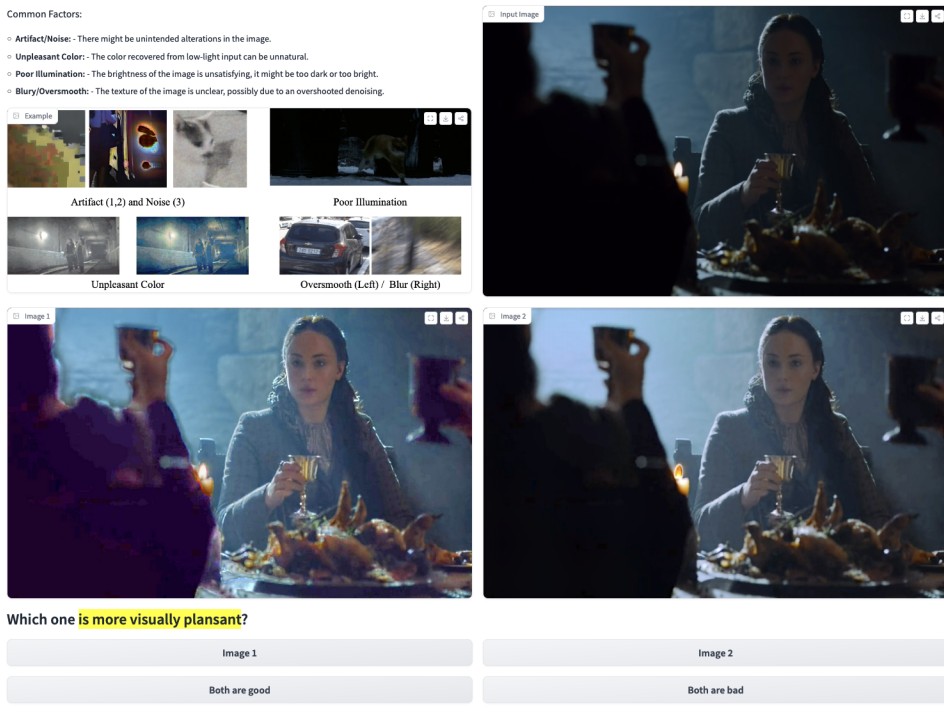

Figure 8: A screenshot of our online user survey system.

- Under Exposure. Gamma correction with $\gamma = 1.5, 2$
- Over Exposure. Gamma correction with $\gamma = 0.75, 0.5$
- Original Exposure. (Gamma correction with $\gamma = 1$)

2. Degradation

- Gaussian Blur with $\sigma_s = 0.1, 0.2, 0.4, 0.8, 1.6$
- Gaussian Noise under level $5, 10, 15, 20, 25$
- Impulse Noise, amount $= 0.01, 0.025, 0.05, 0.075, 0.1$
- Shot Noise under level $60, 45, 30, 20, 12$
- Brightness distortion. First, convert the image into HSV color space, then add $0.1, 0.2, 0.3, 0.4, 0.5$ to $V$.
- Saturate distortion. First, convert the image into HSV color space, then scale component $S$ with $\alpha S + \beta$, where $\alpha = 0.3, 0.1, 2, 5, 20$ and $\beta = 0, 0, 0, 0.1, 0.2$

For every image, we first select a degradation and then perform an exposure shift. In this way, we generate 150 distorted datasets for correlation analysis.

## A.3 TRAINING RECIPE OF THE DETECTORS

In Tab. 1, we train the medium version of YOLOX (Ge et al., 2021) models for "Enhance First" and "Augmentation First" experiments. The training recipe comes mostly from YOLOX for fair comparison, although we adjust the maximum per-image learning rate from $0.05/64$ to $0.05/8$ to keep a consistent global learning rate since we switch to a smaller batch size from 64 to 8. The other data augmentation remains the same as YOLOX suggests. We adopt the Nesterov SGD (Sutskever et al., 2013) optimizer with a momentum of 0.9. The learning rate linearly grows from 0 to the maximum learning rate in 10 epochs, then decays to 0.000001 in a cosine annealing manner for 290 epochs, resulting in 300 epochs of training on the ExDark (Loh & Chan, 2019) dataset. We also adopt the exponential moving average (EMA) during training. The data augmentation scheme involves mosaic, MixUp, color jitter in the HSV range, geometric transformation(random horizontal flipping, random steering, and random rotation), and random change of resolution. In the last 30 epochs, mosaic augmentations are removed for a better adaption of the real-world images. During testing, images are rescaled to $640 \times 640$.

## B FURTHER DISCUSSIONS

### B.1 DISCUSSION ON COLOR JITTOR IN PRE-TRAINED DETECTORS

As we discussed color jitter in Sec. 4, the color jitter during training may introduce discrepancies between detection performance and human perception. To investigate what effect it could bring, we retrained a YOLOX-x detector on the COCO dataset, but without color jitter. This results in a slight performance degradation from 51.1 to 50.8 on the validation split of the COCO dataset. The performance of all detectors, except NeRCO, drops, as in Fig. 4.

|  | Dim image | Zero-DCE | Bread | IAT | RetinexFormer | LLFlow | LD |
|---|---|---|---|---|---|---|---|
| w Color Jitter | 37.0 | 36.2 | 37.3 | 36.8 | 36.4 | 35.7 | 37.0 |
| w/o Color Jitter | 35.8 | 35.7 | 36.4 | 36.2 | 35.5 | 34.9 | 36.2 |

|  | LIME | Nerco | PairLIE | Pydiff | QuadPrior | SCI | SNR | Kind |
|---|---|---|---|---|---|---|---|---|
| w Color Jitter | 35.5 | 33.1 | 35.6 | 32.0 | 37.2 | 36.7 | 36.4 | 36.6 |
| w/o Color Jitter | 34.3 | 33.4 | 34.2 | 31.6 | 36.1 | 35.8 | 35.4 | 35.9 |

Table 4: The detection performance wrt. mAP, with and without color jitter.

We then conducted a correlation study to find out what it could affect. The result can be found in Fig. 9. In general, detectors without color jitter adhere better to human preference. The only outlier is IAT, which delivers less visually pleasant, but extremely clear low-light enhancement results that benefit detection. We found that a linear ensemble of the mAP from two detectors yields an even

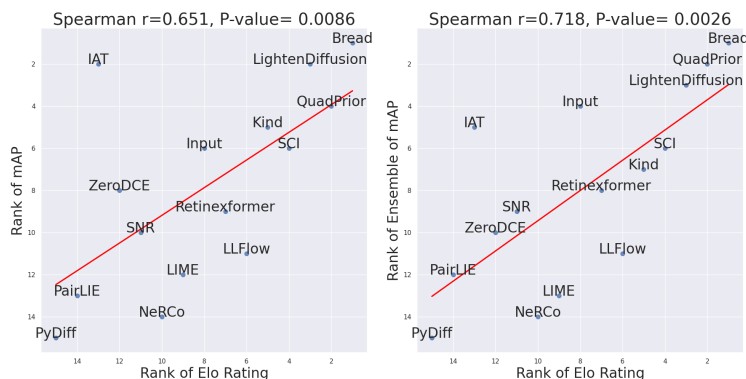

Figure 9: User preference study w/o color jitter and ensemble of two detectors.

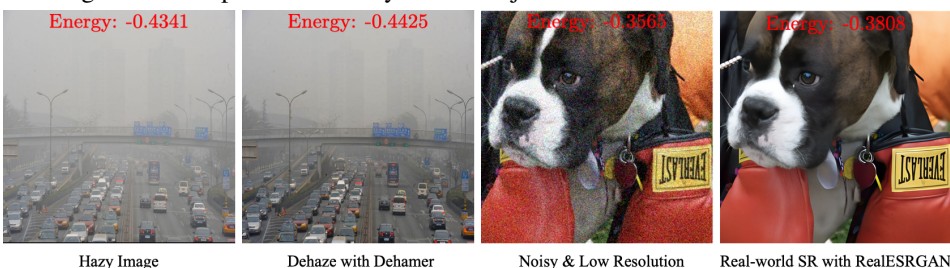

Figure 10: Examples for generalization to other degradations.

better consensus with human perception, raising the correlation from 0.703 to 0.718. This demonstrates that detectors without color jitter adhere better to human preference generally. Ensembling with detectors under various augmentation schemes may further boost the perceptual correlation, which we leave for future work.

### B.2 DISCUSSION ON GENERALIZATION TO OTHER LOW-LEVEL VISION TASKS

To preliminarily show the potential for more restoration/enhancement tasks, due to the short period of rebuttal, we here exhibit cases from dehazing and real-world SR. As depicted in Fig. 10, our method can distinguish clearer results from degraded ones. In fact, we synthesized 8 types of degradation as shown in Fig. 5. These degradations damage the performance of detection and as a result, present a larger energy value. We leave the investigation to more restoration/enhancement tasks for future work.

### B.3 DISCUSSION ON APPLICABILITY TO FACE DETECTION DATASETS AND DETECTORS

Another common choice of low-light detection-based evaluation belongs to face detection on the DARK FACE dataset. To demonstrate the applicability of LIME-Eval on this dataset, we pre-train the detector with the WIDER FACE dataset, a normal-light face detection dataset with 14k images in its training set. We here show qualitative samples in Fig. 11 and their corresponding energy value. It can be observed that clearer, visually pleasant results are assigned with less energy. We also test its generalization ability from face detector to our LIME-Bench dataset, which involves nearly no human faces. Surprisingly, the correlation with user studies drops merely from 0.593 to 0.496, with a P-value of 0.06.

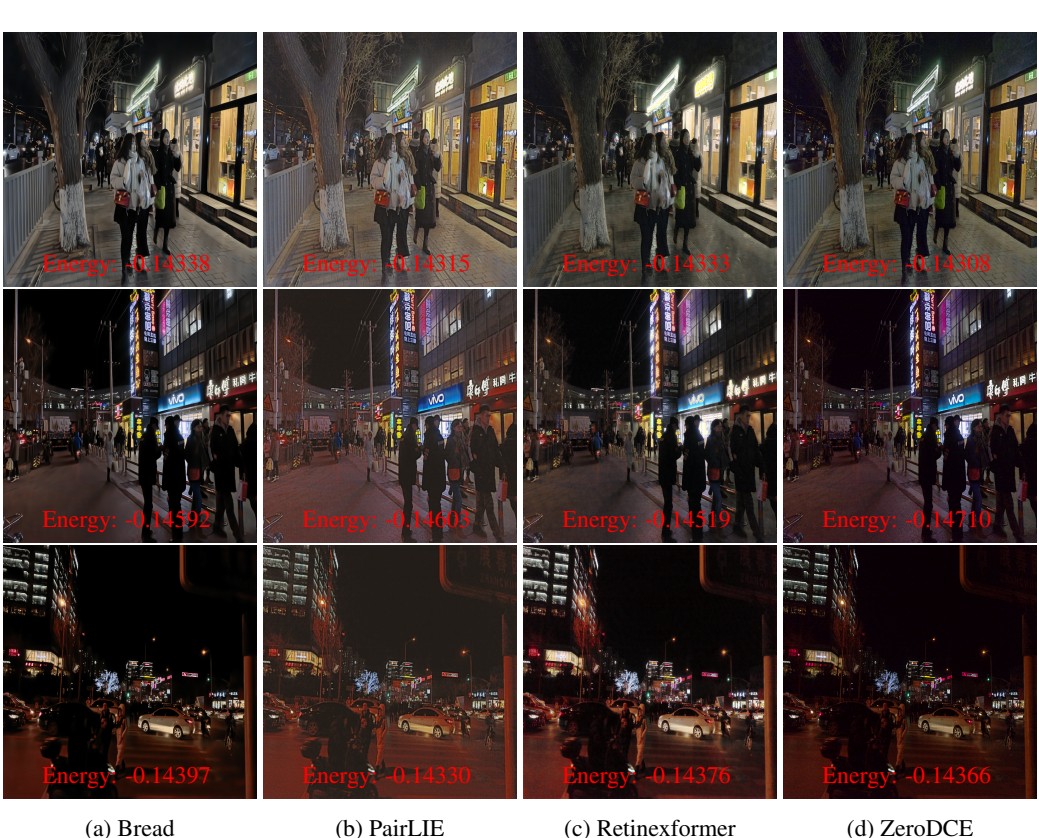

|  (a) Bread | (b) PairLIE | (c) Retinexformer | (d) ZeroDCE |

Figure 11: Examples for face-detector-based LIME-Eval.

## C MORE QUALITATIVE COMPARISONS

In this section, we first exhibited more comparison over existing methods on the ExDark dataset in Fig. 12. As can be found in these Bread (Guo & Hu, 2023) presenting superior visual effects in most cases, the IAT (Cui et al., 2022) has the second-best performance where there exist artifacts in some cases. The ZeroDCE (Guo et al., 2020) has a good color restoration performance, but it suffers from unpleasant noise due to its non-denosing nature. The outputs of RetinexFormer (Cai et al., 2023b) have artifacts in multiple cases.

Qualitative results for models equipped with our energy function are presented in Fig.13. The model equipped with the energy function method can produce more natural outputs. However, as shown in Fig.14, even equipped with our energy loss function, the model still failed to remove severe artifacts in some cases, including persistent checkerboard artifacts in the sky area, and the tendency for pixels in over-exposed areas to be out-of-bounds. Investigating this phenomenon and developing more sophisticated measures to alleviate it is out of the scope of this paper. Yet the case still demonstrates our ability to adjust images to a more natural exposure level.

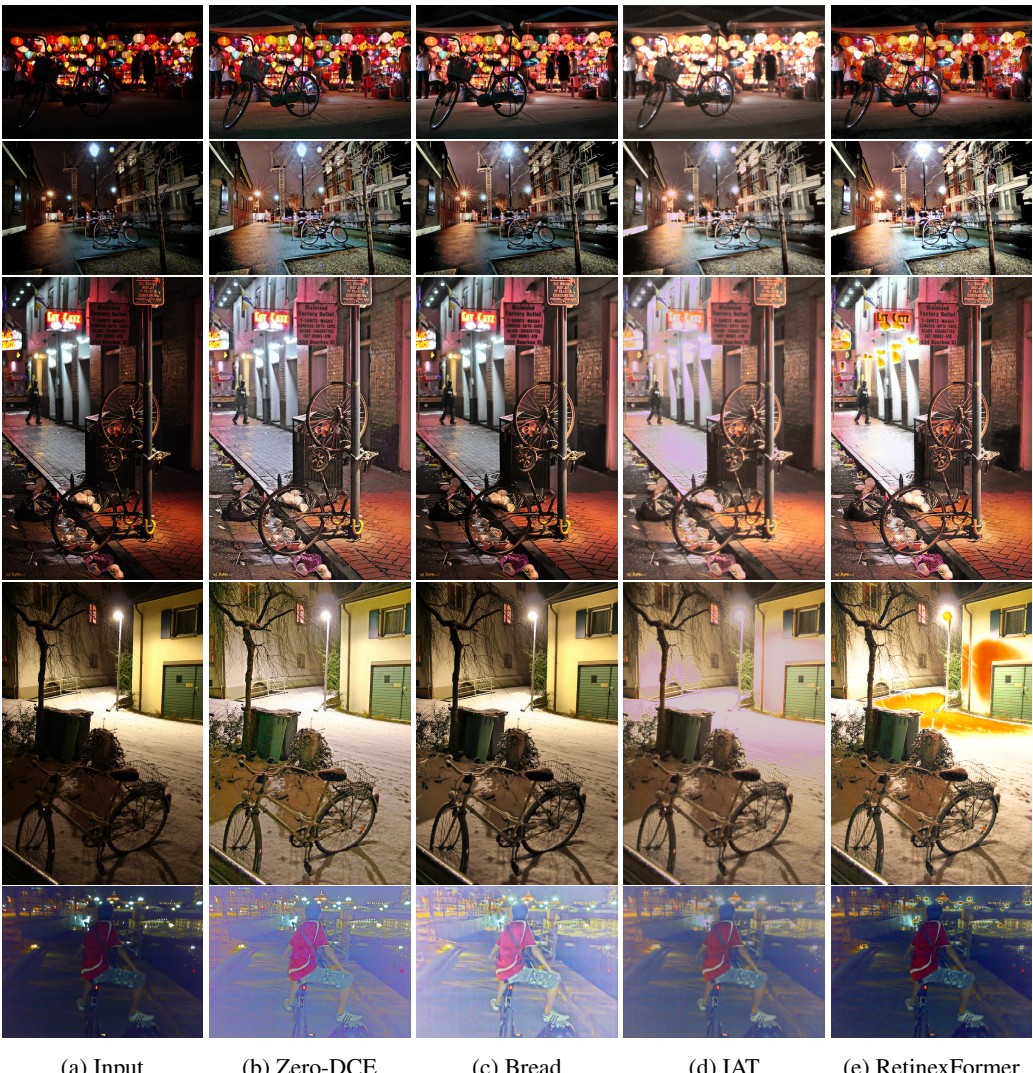

|           |             |          |        |                 |
|-----------|-------------|----------|--------|-----------------|
| (a) Input | (b) Zero-DCE | (c) Bread | (d) IAT | (e) RetinexFormer |

Figure 12: Qualitative comparison on ExDark dataset. Please zoom in for more details.

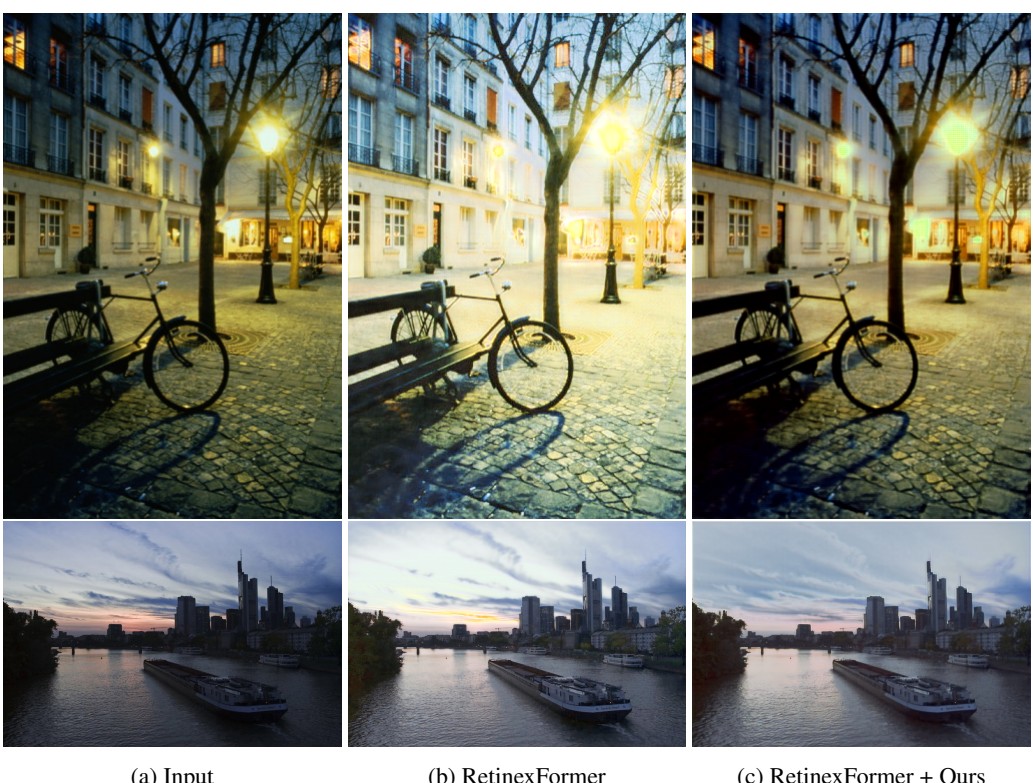

(a) Input          (b) RetinexFormer          (c) RetinexFormer + Ours

Figure 13: Qualitative comparison on ExDark dataset. Please zoom in for more details.

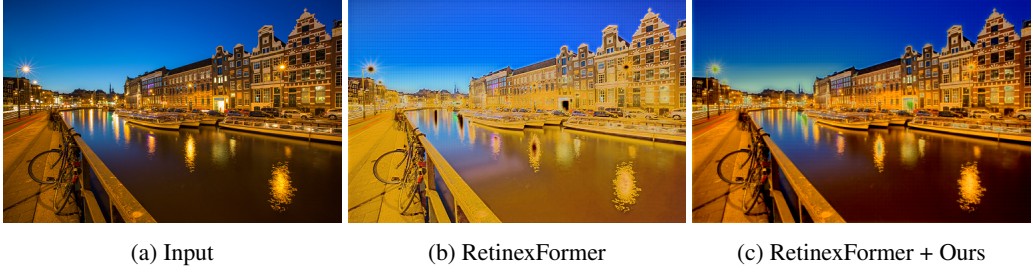

(a) Input          (b) RetinexFormer          (c) RetinexFormer + Ours

Figure 14: A Failure case on ExDark dataset. The model still failed to remove severe artifacts in some cases, even if our energy loss function is adopted.

