# OpenReview forum: "LIME-Eval: Rethinking Low-light Image Enhancement Evaluation via Object Detection"
_ICLR.cc/2025/Conference — ICLR 2025 Conference Withdrawn Submission_

### Official Review · Reviewer_Wgk4 · 2024-10-26

**Soundness:** 2
**Presentation:** 3
**Contribution:** 3
**Rating:** 6
**Confidence:** 4

**Summary:**

This paper reconsiders the evaluation of enhancement quality in low-light tasks, identifies the shortcomings of previous enhancement quality assessment methods, proposes a new dataset based on human preferences, and introduces a novel unlabeled image quality assessment method for enhanced images. The practical significance of the issues addressed in this study is commendable. However, the analysis of the problems is somewhat lacking, and the experimental section could be further improved, as detailed in the Weaknesses section.

**Strengths:**

This paper provides a systematic analysis of methods for measuring the quality of enhanced images in low-light enhancement tasks using object detection. It identifies the shortcomings of previous methods and introduces a new low-light dataset based on human preferences, analyzing the correlation between the evaluations of the object detector and human preferences. Finally, the authors propose a novel unlabeled image quality assessment method that addresses a significant challenge in the current field of low-light enhancement. Overall, this study holds substantial research significance in the domain.

**Weaknesses:**

1. In Chapter 3, the introduction of the need to verify the issue of information loss due to quantization seems abrupt. The earlier sections do not mention the implementation of quantization strategies during the detector's testing process. If quantization was indeed utilized, please provide a brief overview of the quantization process.
2. In Chapter 5, the author fails to analyze how to achieve unlabeled evaluation metrics. The methods section merely outlines the computational processes related to detection-oriented energy-based modeling. The transition from problem introduction to method description is rather abrupt and lacks an analysis of this issue, such as: why is this approach capable of evaluating the quality of image enhancement in an unlabeled manner? Why is the image with the lowest energy value considered optimal? The author should provide relevant analytical descriptions.
3. It is necessary to standardize the meaning of variable x_bg. Why is it referred to as "object output" in line 402, while in line 415 it denotes "background prediction"? Additionally, the method of obtaining x_bg should be specified in this paper. If the network produces multiple object outputs, how should we derive x_bg?
4. In Chapter 5.2, Table 2 presents a comparison of the Spearman correlation coefficients for only IQA algorithm, which is insufficient. I recommend including Spearman correlation coefficients for several additional algorithms to provide a more convincing conclusion.
5. In Chapter 5.3, Table 3 presents the improvements of the method on the object detection metrics mAP and AP50. The author should also indicate whether there are improvements in other metrics (such as recall mAR and AR50). Furthermore, the author could include experimental analyses of additional object detection algorithms (e.g., YOLOX) to enhance the persuasiveness of the conclusions.
6. The article lacks citations for certain terms and methods, such as YOLOX, LOL dataset, and LPIPS. The author should include the relevant references in the text and check for any other missing citations.
7. This paper only discusses the application of detection-oriented energy-based modeling in low-light enhancement tasks. Is this evaluation strategy also applicable to other enhancement tasks (such as rain removal, fog removal, deblurring, etc.)? If possible, I suggest that the authors include a section reflecting on this aspect, such as adding it to the future work section.

**Questions:**

The authors need to solve the questions mentioned, particularly question 2, in the weakness part.

---

> ### Author Response · Authors · 2024-11-24
>
> We deeply appreciate the time and effort you invested in reviewing our manuscript and your positive feedback on our work. Your comments are invaluable, and we have thoroughly addressed them to improve the quality of our paper. Below, we present our detailed responses to your concerns.
>
> **Q1. Quantization.**
>
> The quantization process described in Sec. 3 involves converting the output of an enhancer into unsigned integers within the range of 0-255. For the "Enhancement first" scheme, we pre-enhanced all outputs and saved the enhanced images before training. As a result, the value of inputs to enhancers is discretized into 256 distinct choices. In contrast, the "Augmentation First" scheme uses the intermediate output of enhancers, which remains continuous. To ensure this difference has minimal impact on the results, we conducted an experiment referred to as "Bread-round". In the rebuttal revision, we have provided a clearer explanation of this process.
>
> **Q2. Rationale of energy modeling, and more details on background prediction and object prediction.**
>
> Previous studies [1,2] have demonstrated that classifiers can be interpreted as Energy-Based Models (EBMs), exhibiting an intuitive property: correctly classified samples are associated with lower energy values, whereas misclassified ones are assigned higher energies. Leveraging this insight, we employ energy-based modeling to intimate the accuracy of object detection systems.
>
> In an object detector, we observe that the correctly classified areas are sparse, only with a minimum difference among clear and damaged images, but the logits in those less confident areas are more sensitive to degradations. Modern object detectors, particularly one-stage models (and to some extent, two-stage detectors), typically feature output layers composed of dual classification heads: one for object classification and the other for background classification (there also exists a localization head; however, it addresses location regression rather than classification).  Within the detection pipeline, the model determines whether a data point corresponds to a valid object using the background classification score, which ranges from [0, 1]. A higher score indicates greater confidence in treating the data point as part of the foreground. Subsequently, the class with the highest probability is selected as the final classification output.
>
> To address regions characterized by lower confidence in the background score, we propose a reweighting strategy that amplifies the influence of the classification score in these areas. Specifically, we apply the formulation $(1 - x_{bg}) \cdot x_{cls}$, where $x_{bg}$ represents the background score and $x_{cls}$ denotes the classification score. As this adjustment reduces the overall score magnitude, we introduce a square-root transformation to balance the scaling, yielding a refined confidence measure expressed as $\sqrt{(1 - x_{bg}) \cdot x_{cls}}$. We have refined the manuscript for a clearer presentation of our method.
>
> **Q3. Additional algorithm than IQA for a more convincing conclusion in Chapter 5.2.**
>
> Thanks for your advice. We reorganize the experiments and bring more non-reference metrics for evaluation in Table 2. In the revised version, we involve 5 image quality assessment methods (BRISQUE, NIQE, LIQE, MUSIQ, and ClipIQA), an image aesthetic assessment method NIMA, a color assessment method DeT, and a lightness assessment method LOE to make a more convincing conclusion.
>
> **Q4. Report experiments on more detectors/evaluation metrics in Chapter 5.3.**
> Thanks for your precious advice. We reorganized the experiments in Table 3 and adopted another traditional detector YOLOv8, and an open-vocabulary detector YOLO-World for a direct evaluation.
>
> **Q5. References to certain terms and methods.**
>
> Thanks for your suggestion. In the rebuttal version, we have done thorough proofreading and added related references to terms/methods.

---

> ### Author Response · Authors · 2024-11-24
> **Official Comment by Authors (continued)**
>
> **Q6. Evaluation of more tasks**
>
> Our method does have the potential for more restoration/enhancement tasks.  In fact, we synthesized 8 types of degradation. (Figure 5 exhibits an example of this), which include noise, blurriness, color bias, and exposure shift. These degradations damage the performance of detection and as a result, present a larger energy value. To preliminarily see the potential for more restoration/enhancement tasks, due to the short period of rebuttal, we here exhibit cases from dehazing and real-world SR in Fig.10. From the figure, one can find that our method can distinguish clearer results from degraded ones. We leave a detailed investigation to more restoration/enhancement tasks for future work. We added an analysis of this in the appendix and revised the conclusion/future work part as well.
>
> We hope our responses can address your concerns. Please don't hesitate to respond if you have further concerns.
>
> References:
> [1] W. Grathwohl et al., Your classifier is secretly an energy-based model and you should treat it like one. In ICLR 2020
> [2] R. Peng et al., Energy-based Automated Model Evaluation, In ICLR 2024

---

> > ### Comment · Reviewer_Wgk4 · 2024-11-26
> >
> > Thanks to the authors' careful responses. My concerns regarding most aspects of the paper have been addressed. However, I still have concerns about the innovativeness of the unsupervised evaluation system. Therefore, I have decided to maintain my rating.

---

> > > ### Author Response · Authors · 2024-11-27
> > >
> > > Thanks again for the effort in reviewing our manuscript, and the constructive comments you have made for it!

---

### Official Review · Reviewer_nSif · 2024-11-01

**Soundness:** 1
**Presentation:** 2
**Contribution:** 1
**Rating:** 3
**Confidence:** 5

**Summary:**

The paper argues that using object detectors trained on enhanced low-light images for evaluation is prone to overfitting, reducing reliability. To address this, the authors introduce LIME-Bench, an online platform collecting human preferences to validate the link between human perception and automated metrics. They also present LIME-Eval, a framework using pre-trained detectors without annotations to evaluate image quality, avoiding retraining biases and dependency on dim image annotations.

**Strengths:**

In the background of low-light enhancement, the quality of the enhanced image may indeed be independent of the target detection performance. To verify this phenomenon, they established an online benchmarking platform, LIME - Bench, and developed a label-free assessment metric, LIME-Eval, for quality evaluation of image enhancement. This is a metric that considers both human vision and machine vision.

**Weaknesses:**

I think the experiment lacks conviction.
1. Qualitative comparisons on samples from the ExDark dataset are inadequate and even wrong in my opinion. The reason for this is that the augmentation strategy is random and not a fair input to the target detector.
2. How is LIME-Eval integrated into Retinexformer? I don't see it in the appendix either, rather than you just stating the training details.
3. If LIME-Eva is integrated into the detection, will it participate in the calculation of the parameter gradient of the neural network? If it is involved, then I think that LIME-Eva just has the efficacy of a loss function, since the low-light enhanced image is then fed to the target detector for training, which is not referential in nature, as you conclude in Table I.

**Questions:**

See Weaknesses.

---

> ### Author Response · Authors · 2024-11-24
>
> Thanks for the time and effort spent on our work.
>
> There seems to be some misunderstanding, thus we would like to clarify the contribution of our work in this response.
>
> Our work begins with a preliminary experiment to investigate which evaluation scheme fits best for human perceptual. The result shows that directly evaluating enhanced low-light images with normal-light pre-trained enhancers gives a better understanding of human perceptual, rather than training on the enhanced images.
>
> To confirm the advantage of this method, we add more low-light enhancement methods, launch the online benchmark platform, LIME-bench, to collect pairwise user preference, and most importantly, come up with the conclusion that the direct evaluation scheme does have a strong correlation to human preference. We also discuss the factor that forms discrepancies between machine and human preference.
>
> Finally, we proposed a label-free and reference-free metric for evaluating low-light enhancers, namely LIME-eval. Since it requires no detection labels, it can be used on natural images, not limited to annotated detection datasets.
>
> We highlighted the contribution of our work to avoid possible confusion, on the pipeline and technical contribution of this manuscript in the rebuttal revision.
>
> We respond to specific concerns as follows:
>
> **Q1. Augmentation for Qualitative comparisons on ExDark**
>
> We agree with the opinion that augmented images are not legitimate inputs for qualitative comparison. However, we do NOT use augmentation on any qualitative samples of ExDark, the input is fed as is in all of the qualitative samples. By posing these images, we would like to give a qualitative view of how this enhancer performs in real-world samples.
>
> **Q2. How is LIME-Eval integrated into Retinexformer?**
>
> As a loss term. Since it requires no object/ground-truth annotation, we just train Retinexformer with their released codebase with exactly the same data they used, but add LIME-eval as an additional loss term.
>
> **Q3: Concerns for training with target detector.**
>
> Please note that we are not discussing retraining enhancers, but object detectors in Table 1. Since the retrained Retinexformer has never been back-propagated with detection labels or images from the detection dataset, it meets no contradiction with what we have stated in Section 3.
>
> We hope our responses can address your concerns. Please don't hesitate to respond if you have further concerns.

---

> > ### Author Response · Authors · 2024-11-27
> >
> > Dear Reviewer nSif,
> >
> > We sincerely thank you for taking the time to review our paper. We have posted our point-to-point responses above. Since the public discussion phase is ending very soon, it would be appreciated if you read our responses and let us know your feedback. Please don't hesitate to respond if you have further concerns.
> >
> > Sincerely,
> > Authors of ICLR Submission1464

---

### Official Review · Reviewer_egde · 2024-11-02

**Soundness:** 3
**Presentation:** 3
**Contribution:** 3
**Rating:** 8
**Confidence:** 4

**Summary:**

This paper addresses the challenge of evaluating low-light image enhancement without paired ground-truth data, as current high-level vision task evaluations can often lead to overfitting and unreliable metrics. To improve this, the authors introduce LIME-Bench, an online platform that collects human preference data for low-light enhancement, creating a valuable dataset to correlate human perception with automated metrics. They further propose LIME-Eval, a novel evaluation framework using pre-trained detectors without retraining or annotation dependencies, to assess enhanced image quality. Through energy-based confidence map assessment, LIME-Eval effectively gauges enhancement accuracy. Extensive experiments validate LIME-Eval's effectiveness, and the code will be publicly available.

**Strengths:**

1. this paper addresses the challenges of evaluating low-light enhancement techniques via the proposed LIME-Bench online platform, collecting human preference low-light images for low-light enhancement techniques, and the LIME-Eval framework without fine-tuned detection models and annotation labels.

2. the paper demonstrates the uncorrelation of high detection performance to the enhancement quality of the enhanced images through comprehensive studies and analysis. This provides a scientific foundation to support their proposed LIME-Bench platform and LIME-Eval framework.

3. the paper addresses the challenge of evaluating low-light enhancement techniques without fine-tuning detection models and acquiring annotation labels through the energy-based modeling technique.

**Weaknesses:**

1. the purpose of consdering background prediction in the LIME-Eval framework is vauge. It would be great for authors to describe the benefit of integrating background prediction into the energy-based evaluation function.

**Questions:**

Please refer to Weaknesses.

---

> ### Author Response · Authors · 2024-11-24
>
> We would like to express our gratitude for the time and effort spent on reviewing our manuscript and for your appreciation of our work. Your comments are valuable for us, and we have accordingly revised the manuscript in Sec. 4. of the rebuttal revision. Please find the response to the concern below.
>
> **Q: Explain background prediction.**
>
> For modern object detectors, particularly one-stage detectors (and to some extent, two-stage detectors), feature output layers are typically composed of dual classification heads: one for object classification and the other for background classification (there also exists a localization head; however, it addresses location regression rather than classification). Within the detection pipeline, the model determines whether a data point corresponds to a valid object using the background classification score, which ranges from [0, 1]. A higher score indicates greater confidence in treating the data point as part of an object. Subsequently, the class with the highest logit is selected as the final classification output.
>
> A neat property of energy-based methods is that correctly classified samples are associated with lower energy values, whereas misclassified ones are of higher energies. In an object detector, we observe that the correctly classified areas are sparse, only with a minimum difference among clear and damaged images, but the logits in those less confident areas are more sensitive to degradations. To address regions characterized by lower confidence in the background score, we propose a re-weighting strategy that amplifies the influence of the classification score in these areas. Specifically, we apply the formulation $(1 - x_{bg}) \cdot x_{cls}$, where $x_{bg}$ represents the background score and $x_{cls}$ denotes the classification score. As this adjustment reduces the overall score magnitude, we introduce a square-root transformation to balance the scaling, yielding a refined confidence measure expressed as $\sqrt{(1 - x_{bg}) \cdot x_{cls}}$.
>
> We hope our responses can address your concerns. Please don't hesitate to respond if you have further concerns.

---

> > ### Comment · Reviewer_egde · 2024-11-26
> >
> > Thank you for author's response, which addressed my questions for this paper. I will keep my original rating.

---

> > > ### Author Response · Authors · 2024-11-27
> > >
> > > Thanks again for the time and effort you spent on our work!

---

### Official Review · Reviewer_3LRc · 2024-11-03

**Soundness:** 3
**Presentation:** 3
**Contribution:** 3
**Rating:** 8
**Confidence:** 4

**Summary:**

The work: LIME-Eval: Rethinking Low-Light Image Enhancement Evaluation via Object Detection, critiques the standard practice of evaluating low-light image enhancement by retraining object detectors on enhanced images—a method prone to overfitting and reduced reliability. Instead, the authors propose LIME-Eval, an energy-based evaluation framework that eliminates the need for retraining detectors or relying on annotated data. They also introduce LIME-Bench, an online platform that collects human preferences on low-light enhancement to align automated metrics with human perception. Experiments on datasets like ExDark demonstrate LIME-Eval’s effectiveness in evaluating enhanced images in a way that closely mirrors human judgment.

**Strengths:**

1. Higher detection accuracy does not necessarily correlate with superior enhancement quality, which aligns with my understanding of low-light enhancement models. There exists a gap between low-level, human-visual-oriented perception and high-level, machine-vision-oriented objectives. Additionally, as noted in line 68, 'Is fine-tuning detectors a valid approach to evaluating enhancers?' this raises a valuable point. Relying on fine-tuning detectors to fully assess low-light detection performance can lead to overfitting and is not necessarily a scientifically rigorous approach.

2. The paper is well written, easy to understand, and authors conduct extensive experiments to validate their approach, offering thorough insights into the correlations between detector performance, human ratings, and LIME-Eval metrics.

3. LIME-Bench provides a structured way to incorporate human preferences into evaluation, bridging the gap between machine-based metrics and human perceptual quality. Which could also be helpful to future low-light vision research.

**Weaknesses:**

1. In the 'enhanced before data augmentation' setting (Table.1), is there a process like 'color jitter'? I think 'color jitter' could introduce significant interference, as it might substantially degrade the results of low-light enhancement if applied. Also the normalization of image maybe also affect the low-light enhancement models. Could the authors provide more detail on this part?

2. The paper mainly uses ExDark and synthetic datasets, can this evaluation tool also been used in the low-light face detection dataset UG2+ DARK FACE ?

3. Some previous low-light object detection works, although have not used pre-defined enhancement modules, they remain closely related to this study. They did not use image enhancement models but instead employed other approaches, such as self-supervised learning and domain adaptation, to address the issue of low-light object detection. It would be beneficial for the authors to reference and discuss these works in the related works section (e.g., [1, 2, 3]).

[1]. Multitask AET with Orthogonal Tangent Regularity for Dark Object Detection, ICCV 2021

[2]. Similarity min-max: Zero-shot day-night domain adaptation, ICCV 2023

[3]. Boosting Object Detection with Zero-Shot Day-Night Domain Adaptation, CVPR 2024

**Questions:**

See weakness.

---

> ### Author Response · Authors · 2024-11-24
>
> We sincerely appreciate the time and effort you dedicated to reviewing our manuscript. We are particularly delighted by the insightful nature of your comments, which have greatly inspired us to refine and improve our work in the rebuttal revision. Please find the response to the concern below.
>
> **Q1: What is the training recipe for the detectors? Does the color jitter and normalization apply to the models?**
>
> ImageNet normalization is not applied during training. As suggested by the authors of YOLOX, we adopt a simple division of 255.0.
> Since we adopt pre-trained YOLOX models on the COCO dataset for direct evaluation, we align the training recipe of YOLOX to keep consistency during comparisons.  The data augmentation pipeline of YOLOX involves two stages:
> In the first stage (270 epochs), mosaic, MixUp, color jitter in HSV color space, and geometric augmentations are used, while in the second stage (30 epochs), mosaic is turned off to make a better alignment to the real-world images. To make a clearer view of the training setting of the models in Table 1, We have added further training details of the detectors to Appendix A.2.2 in the rebuttal revision.
>
> To investigate what effect it could bring, we retrained a YOLOX-x detector on the COCO dataset, but without color jitter. This results in a slight performance degradation from 51.1 to 50.8 on the validation split of the COCO dataset. The performance of all detectors, except NeRCO, drops.
>
> | | Dim image | Zero-DCE | Bread | IAT | RetinexFormer | LLFlow | LD | LIME | Nerco | PairLIE | Pydiff | QuadPrior | SCI | SNR | Kind |
> |---|---|---|---|---|--|---|---|---|---|---|---|---|---|---|---|
> |w Color Jitter  | 37.0 | 36.2 | 37.3 | 36.8 | 36.4 | 35.7 | 37.0 |  35.5 |33.1 | 35.6 | 32.0 | 37.2 | 36.7 |36.4 | 36.6 |
> | w/o Color Jitter | 35.8 | 35.7  | 36.4  | 36.2 | 35.5 | 34.9 | 36.2 | 34.3 | 33.4 | 34.2 | 31.6  | 36.1 | 35.8 | 35.4 | 35.9|
>
> We then conducted a correlation study to investigate what it could affect, as in Appendix B.2 of the rebuttal revision. In general, detectors without color jitter adhere better to human preference. The only outlier is IAT, which delivers less visually pleasant, but extremely clear low-light enhancement results that benefit detection. We found that a linear ensemble of the mAP from two detectors (0.25 * w/o color jitter + 0.75 * color jitter) yields an even better consensus with human perception, raising the correlation from 0.703 to 0.718. This demonstrates that detectors without color jitter adhere better to human preference generally. Ensembling with detectors under various augmentation schemes may further boost the perceptual correlation, which we leaves for future work.
>
> **Q2: Evaluation with face detectors.**
>
> It is positive that our method can be applied to other detection datasets, such as the Dark Face dataset, as the reviewer mentioned. To demonstrate this, we have pre-trained the model with WIDER FACE and tested the model with enhanced DARK FACE results. Due to the short period of the discussion phase, we exhibit visual results for the dark in Appendix B.3 of the rebuttal revision. Effectively, though, it is limited to face images while facing challenges to evaluate natural images without human faces. To see this, we evaluate our face detector with our LIME-Bench dataset, which contains few human faces. The correlation drops from 0.593 to 0.496, demonstrating its limitation in evaluating images in the wild.  Learning on the ExDark dataset leads to an awareness of objections to the model, which can be generalized better in real-world scenarios. In future work, we are planning to evaluate with open-vocabulary detectors, which can detect arbitrary objects. Thanks again for your insightful comments.
>
> **Q3: Add related work to Low-light object detection without enhancement.**
>
> Thanks for your suggestion. In the revision, we have added a paragraph in the Related Work section to describe representative existing schemes in low-light object detection without enhancement.

---

> > ### Comment · Reviewer_3LRc · 2024-11-26
> >
> > Thank you for the detailed response, which addressed many of my questions. I'll maintain my original rating.

---

> > > ### Author Response · Authors · 2024-11-27
> > >
> > > Thanks again for your time and effort spent on our work!

---

### Author Response · Authors · 2024-11-24
**Notes for our rebuttal revision**

Dear ACs and Reviewers,

We would like to thank you for your time and effort spent on our manuscript. The concerns and suggestions are valuable for us to further improve the quality of this work. We have taken these comments on board to make the rebuttal revision.

The following summarizes the major points of our rebuttal:

+ We enriched the Related Work Section, by adding the discussion on low-light object detection schemes, thanks to Reviewer 3LRc's valuable advice.

+ We refined Section 4 and added more explanation to our energy-based modeling framework, as suggested by Reviewer Wgk4 and egde.

+ We added additional competitors in Table 2, and more evaluation metrics in Table 3 to make comparisons more comprehensive, thanks to Reviewer Wgk4's kind reminder.

+ We detailed the training recipe for the detectors in the Appendix A.3 as suggested by Reviewer 3LRc.

+ We added a discussion on the applicability of our work on face detection, and generalization potential to other low-level vision tasks in Appendix B, inspired by Reviewers 3LRc and Wgk4.

+ We highlighted the contribution of our work to avoid possible confusion, as mentioned by Reviewer nSif, on the pipeline and technical contribution of this manuscript.

+ We performed thorough proofreading to eliminate typos and mistakes in the previous version. We also added appropriate references, as suggested by Reviewer Wgk4.

We have provided detailed responses to the reviewers. We sincerely hope that our response can address the concerns.

Thanks again for your review.

The authors of LIME-Eval

---

### Note · Authors · 2025-03-08

I have read and agree with the venue's withdrawal policy on behalf of myself and my co-authors.

---

### Meta-Review · Area_Chair_cWV8 · 2024-12-20

**Metareview:**

The paper addresses the challenges in evaluating low-light image enhancement due to the lack of ground-truth references. It critiques a widely-used evaluation approach, which measures object detection accuracy on enhanced low-light images, arguing that it is prone to overfitting and unreliable. To address this, the authors propose a LIME-Bench, an online benchmarking platform designed to gather human preferences for low-light image enhancement. Besides, they propose a novel evaluation framework, LIME-Eval, that uses pre-trained object detectors without requiring annotations.

This paper receives significant dispersed scores among the reviews. The two reviewers, 3LRc and egde, largely appreciate the paper's contributions, particularly its emphasis on highlighting the distinction between low-light enhancement and detection. Besides, they all think this paper is well-written and well-organized.

The major negative comments come from Reviewer nSif, concerning center on the adequacy of the experiments, particularly the fairness of the augmentation strategy, the role of LIME-Eval in detection, and its integration with Retinexformer, which lacks clarity and rigor. These issues indicate that the methodology and experimental validation require further refinement, making the paper unsuitable for acceptance at this stage.

**Additional Comments On Reviewer Discussion:**

Reviewer nSif's concerns center on the adequacy of the experiments, particularly the fairness of the augmentation strategy, the role of LIME-Eval in detection, and its integration with Retinexformer, which lacks clarity and rigor. The authors' rebuttal addressed most of these concerns point-to-point, providing additional clarification. However, despite these efforts, the reviewer nSif ultimately keeps the original score.

---

### Decision · Program_Chairs · 2025-01-22

Reject